



# Reduced seed-material dependence of a condensation particle counter

Peter J. Wlasits[1,2], Joonas Enroth[3], Joonas Vanhanen[3], Aki Pajunoja[3], Hinrich Grothe[1], Paul M. Winkler[2], Dominik Stolzenburg[1]

[1]Institute for Materials Chemistry, TU Wien, Vienna, 1060, Austria
[2]Faculty of Physics, University of Vienna, Vienna, 1090, Austria
[3]Airmodus Ltd., Helsinki, 00560, Finland

*Correspondence to*: Dominik Stolzenburg (dominik.stolzenburg@tuwien.ac.at)

**Abstract.** Modern condensation particle counters (CPCs) are indispensable instruments for studies of aerosols in all measurement environments. Relying on heterogeneous nucleation as basic principle, the composition dependence of particle activation is a source of profound uncertainty for the accurate assessment of particle number concentrations. While
development efforts successfully pushed down minimum detectable particle sizes in recent years, composition-dependent counting efficiencies have remained to be a persisting issue in aerosol research. Addressing this pressing problem, we present calibrations of a newly developed CPC, the Airmodus A30, that uses non-hazardous propylene glycol as working fluid. Our results conclusively demonstrate that composition-dependent particle detection can be reduced to the brink of disappearance
by innovative choice of the working fluid. Counting efficiencies were determined for a set of size-selected and chemically diverse seed particles and the measured 50 % cutoff diameters were compared to previous studies. Using computational fluid dynamics simulations, we show that the composition dependence appears to decrease with increasing saturation ratios achieved inside the CPC. Hence, our study assists in the development of future CPCs and elucidates a potential mechanism to reduce measurement uncertainties arising from composition-dependent counting efficiencies.


## 1 Introduction

Condensation particle counters (CPC) are widely used aerosol detectors to provide a measurement of the total particle number concentration in environmental monitoring, in automotive exhaust testing or in scanning mobility particle spectrometers (SMPS), where the concentration information is used to obtain the particle number size distribution below 1 µm. The clear
advantage of CPCs is their ability to provide single particle counting with detection of particles down to the cluster-particle transition regime of a few nanometers. While aerosol particles with sizes well below 200 nm are difficult to detect by optical means (at least using visible light), CPCs artificially create an environment of a supersaturated vapor, which condenses onto the nanoscale aerosol particles and thereby grows them to optically detectable sizes. However, this condensation process initiated by heterogenous nucleation of the supersaturated vapor on the aerosol surface is a complex interaction between the
aerosol seed and the supersaturated vapor (Winkler et al., 2008), which typically leads to a minimum detectable size. For particles smaller than the minimum detectable size, the achieved supersaturation is not sufficient to overcome the Kelvin barrier due to the increased curvature of the particle seed's surface. CPCs are therefore characterized by their counting



efficiency (cutoff) curve, where the detected concentration of a certain seed particle size is compared to a reference instrument, typically a Faraday Cup Electrometer (FCE), which is commonly expected to have unit detection efficiency even for the

smallest clusters but cannot provide single-particle counting.

The shape of these cutoff curves and the diameter, where 50 % activation is achieved ($d_{50}$ in the following), depend on the complex interactions between the chemical composition, the charging state of the sampled aerosol particles and the properties of the used vapor (Kangasluoma et al., 2013; 2014) as well as on the flow rates and overall instrument design. Over the last decades several novel condensation-type particle detectors have been developed using different working fluids mainly to push

the limits of the $d_{50}$ value towards smaller sizes: 1) using n-butanol ($C_4H_{10}O$), which is chemically resistant and by usage of capillary design it can even activate particles as small as 2 nm (Stolzenburg and McMurry, 1991); 2) using water ($H_2O$), which is environmentally friendly, odorless as well as easy to handle and can also be boosted towards sub-3 nm detection (Hering et al., 2017; 2019); 3) using diethylene glycol ($C_4H_{10}O_3$, DEG), which has a comparatively low vapor pressure and allows activation of molecular clusters well below 2 nm (Vanhanen et al., 2011). However, it was shown recently that for all of these

CPCs a significant seed-material composition dependence of the cutoff curve exists with differences in the $d_{50}$ values of up to 1 nm for the latest generation of ultrafine detectors (Wlasits et al., 2020).

This implies a significant source of uncertainty within the typical applications of CPCs as the chemical composition of the sampled aerosol is often unknown. Kangasluoma and Kontkanen (2017) showed that an uncertainty of ±0.5 nm in the $d_{50}$ value can lead to an increase of uncertainties by a factor of 10 in both the measured total particle number concentration and the

SMPS-derived measured size distribution, especially when narrow size distributions with steep gradients in size are sampled as in the case of almost all systems where new particle formation (NPF) from gas-to-particle conversion takes place (Stolzenburg et al., 2023a). In the case of regional NPF the uncertainty in the total particle number concentration or particle size distribution directly translates into uncertainties of derived quantities such as the particle formation and growth rates (Stolzenburg et al., 2023b). Chamber experiments have thus taken challenging approaches to correct their derived values to

avoid these potentially dramatic uncertainties (Dada et al., 2020; Rörup et al., 2022). These uncertainties are even more important when it comes to vehicular exhaust emissions, where a non-volatile particle number emission limit of $6 \cdot 10^{11}$ particles $km^{-1}$ was introduced into European Euro 5/6 legislation (Giechaskiel et al., 2011). Several studies have shown that changes in the seed material composition also heavily influence these CPC measurements with a nominal cutoff of 23 nm posing an important challenge to achieve intercomparability of measurements within these legislative procedures (Giechaskiel et al.,

2009; Giechaskiel et al., 2011; Terres et al., 2021; Krasa et al., 2023).

The development of CPCs with reduced seed-material composition dependence can thus be considered a milestone in aerosol technology. In the following sections we present conclusive evidence that aforementioned seed-material dependences can be efficiently reduced by using propylene glycol as condensable vapor in a newly developed CPC and that this behavior is related to high saturation ratios achieved in the instrument.




## 2 Characteristics of propylene glycol and the Airmodus A30

Despite all efforts undertaken so far, the composition dependence of activation curves is still posing a significant challenge. However, persisting research efforts yielded a substantial decrease in the minimum detectable size by choosing specific working fluids. Those results hint that activation efficiencies are profoundly influenced by specific interaction between the chemical composition of the seed particle and the molecular constitution of the vapor (Krämer et al., 2000; Keshavarz et al., 2020; Toropainen et al., 2021). Employing the practical criteria for working fluids outlined in Magnusson et al. (2003) and Iida et al. (2009), propylene glycol is a suitable substance since it is non-hazardous and has a dynamic viscosity below 50 mPa·s at room temperature. Chemical and physical properties of propylene glycol in comparison to n-butanol are presented in Table S1. In general, propylene glycol (IUPAC nomenclature: propane-1,2-diol, $C_3H_8O_2$) is a clear and colorless synthetic substance, that, unlike n-butanol, is practically odorless (National Center for Biotechnology Information, 2023). Having a similar molecular weight as n-butanol, propylene glycol has a considerably lower vapor pressure (s. Table S1). Hence, droplet sizes following condensation will be smaller due to a lower collision rate between molecules of the working fluid and the seed particles (Iida et al., 2009; Toropainen et al., 2021) and CPC optics need to be adjusted accordingly. Since a higher surface tension compared to n-butanol (s. Table S1) also increases the height of the energy barrier for gas-to-particle conversion (Iida et al., 2009), propylene glycol-based CPCs must be operated at a higher saturator temperature and a decreased condenser temperature compared to the butanol-based models (s. Table 1). However, using propylene glycol as working fluid is linked to different and particularly favorable seed-vapor interaction which might be connected to the vapor molecules having a higher polarity compared to butanol (similar to that of water), but similar mass as butanol and hence a lower diffusivity compared to water.

The overall design of the Airmodus A30 is presented in Fig. 1. The Airmodus A30 is a laminar flow-type CPC with 0.2 L min⁻¹ sample flow rate and 1.3 L min⁻¹ of additional transport flow. The design has no internal dilution but only a core sampling configuration at the inlet of the CPC. The carrier flow can be switched on and off allowing direct measurement of the sample flow rate from the inlet of the CPC. Excess condensate can be removed by using a condensate removal function to prevent contaminating the saturator.

## 3 Main findings and comparison to previous studies

The experimental approach follows the description presented in Wlasits et al. (2020). A schematic of the experimental setup, including a brief description, is presented in the Supplement (s. Fig. S1). Measurements were conducted using two versions of the newly developed Airmodus A30 (Airmodus Ltd., Helsinki, Finland); i.e. a butanol-based variant and a variant of the A30 using propylene glycol as working fluid, which are completely identical except for the used working fluid and temperature settings in saturator and condenser. Counting efficiencies were determined based on a set of measurements with seeds produced from four different particle materials: silver wool (Ag), sodium chloride (NaCl), ammonium sulfate ((NH₄)₂SO₄) and β-caryophyllene (C₁₅H₂₄). Details of the particle materials in use are presented in Table S2 in the Supplement.



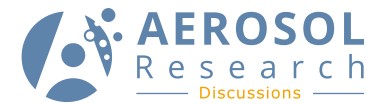

### 3.1 Comparison of working fluids

The first set of experiments aimed at directly comparing the butanol-based A30 variant to the propylene glycol-based variant. The corresponding results are presented in Fig. 2. The figure shows the counting efficiency measurements corresponding to

the aforementioned set of seed particle materials for each of the two working fluids together with fits to guide the eye. The solid lines correspond to calculations of the transmission efficiency depending on particle size based upon diffusion losses as described by Gormley and Kennedy (1949).

Evidently, measurements with the butanol-based CPC unveiled a profound difference between the counting efficiencies of different seed particles (s. Fig. 2a). While the 50 % cutoff diameters of Ag, $C_{15}H_{24}O_x$ and $(NH_4)_2SO_4$ seeds show smaller

differences of up to almost 1 nm and are centered around 6.5 nm, the curve for NaCl seeds is clearly even more detached from the other seeds and located at larger mobility diameters. Furthermore, the observed shift is connected to a notable flattening of the curve's slope. As can be also seen from data presented in Table 1, the 50 % cutoff diameter for NaCl seeds is located at approximately 10 nm. These findings corroborate with the results previously presented for butanol-based CPCs in Wlasits et al. (2020). The same set of measurements was repeated using the propylene glycol-based A30. It is crucial to note that the

corresponding counting efficiencies do not exhibit any significant differences (s. Fig. 2b). These conspicuous discrepancies between the two identically constructed CPCs must be attributed to changes related to the working fluids in use. Unlike the 50 % cutoff diameters corresponding to the use of butanol, the cutoff diameters for any of the seed particles are centered around 5.8 nm in case of the propylene glycol-based CPC. However, counting efficiencies of approximately 0.95 are reached at around 20 nm in both cases.

Figure 2 also allows for comparison of measured data to calculated transmission efficiencies of particles within in the CPC. Diffusional losses along the particles' path from the CPC inlet to the condenser were assessed (solid lines in Fig. 2). These calculations revealed that the measured counting efficiencies cannot solely be attributed to diffusional losses within the instrument, i.e. the propylene glycol-based A30 does not lose its composition dependence just because it achieves unity activation efficiency for all particles. Theoretical transmission efficiencies start agreeing fairly well with measured data for

larger seed particles having diameters of around 9 and 14 nm, for the butanol- and propylene glycol-based CPC respectively. The results of additional calculations outlining the combined effect of theoretical activation curves and expected losses arising from diffusion are presented in Fig. S2 in the Supplement and reproduce the shape of the actually measured counting efficiency curves remarkably well. Diffusional losses with an effective length of 2.6 times the design values would be required to achieve similar diffusion loss-limited cutoffs, but in that case the (design-identical) butanol counting efficiency curves could not have

the same shape as measured and plateau values of the propylene glycol-based A30 should also be smaller (s. Fig. S3).

The observed differences in the detection efficiencies can be related to the supersaturation profiles inside of the CPCs' condensers (Stolzenburg and McMurry, 1991; Tauber et al., 2019a). Computational fluid dynamics (CFD) simulations yielded maximum saturation ratios of 2.56 and 5.00 for the butanol-based and propylene glycol-based A30, respectively.





## 3.2 Effects of decreased saturator temperatures

To test the relation between composition dependence and saturation ratio, we reduced the peak saturation ratios in the propylene glycol-based A30. As shown in previous studies, decreasing the temperature difference between the saturator and the condenser, entails a decrease of peak supersaturation in the condensers of butanol-based CPCs (Giechaskiel et al., 2011; Barmpounis et al., 2018). For that reason, the saturator temperature, $T_{sat}$, was successively decreased while keeping the temperature of the condenser constant (s. Table 1). Measurements of the counting efficiencies using the propylene glycol-based A30 were conducted at saturator temperatures of 41 °C and 37 °C. The results are presented in Fig. 3.

Figure 3a emphasizes that a decrease of $T_{sat}$ by 7 °C has proven to be already sufficient to reinduce a significant composition dependence of the 50 % cutoff diameters. Within uncertainty limits, the counting efficiency remained to be close to 0.95 at 20 nm (s. Fig. 3a). These findings therefore indicate that reduced supersaturation inside of the CPC's condenser is indeed the origin of the material dependences observed in Fig. 2a.

As shown in Fig. 3b, further decrease of $T_{sat}$, consistently yields even more material dependence of the counting efficiencies. The measured curves flatten drastically at a saturator temperature of 37 °C corresponding to a reduction by 11 °C from default saturator settings. The difference between the $d_{50}$ of NaCl and the $d_{50}$ of Ag is now given by approximately 4 nm and differences up to 30 % in counting efficiencies between the different materials are observed close to the $d_{50}$. However, counting efficiencies at 20 nm remained within 5-10 % of 0.7 for all four different seed materials. This possibly indicates a lower overall plateau height, which is potentially caused by droplet sizes being too small in the optics due to non-sufficient absolute propylene glycol vapor concentrations in the CPC. Figure S4 in the Supplement shows that the shift of the counting efficiency curves to larger diameters (as expected from theoretical considerations shown in Fig. S2 in the Supplement) and the flattening of the curves is significantly more pronounced in case of NaCl compared to e.g. Ag.

The presented results clearly underline the importance of seed-vapor interactions during particle activation and lend further support to the main conclusions drawn in Wlasits et al. (2020). We therefore hypothesize that favorable vapor-seed interactions (Li and Hogan Jr., 2017; Wlasits et al., 2020) are primarily responsible for the just marginally different activation of Ag seeds at already reduced saturation ratios.

## 3.3 Relating the 50 % cutoff diameters to the maximum saturation ratios

The maximum difference in any of the four cutoff diameters per instrument and setting, i. e. the difference between the largest cutoff diameter and the smallest cutoff diameter for any seed particle per CPC and setting ($\Delta d_{50,max}$), was used as a proxy for the composition dependence of the counting efficiency. Figure 4 shows $\Delta d_{50,max}$ as a function of the calculated maximum saturation ratios in the condenser of the CPC in use.

Data for this comparison corresponding to the set of butanol-based CPC manufactured by TSI Inc. (Models 3772, 3776 and a tuned version of Model 3776) and by Airmodus Ltd. (Model A20) were taken from Wlasits et al. (2020). The tuned TSI 3776, denoted by superscript "T", was operated at lower temperature settings and with an increased inlet flow rate and has already been found to have no significant composition dependence in its counting efficiency in previous calibration experiments



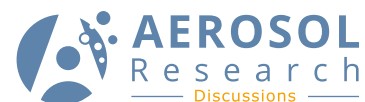

(Tauber et al., 2019b; Brilke et al. 2020; Wlasits et al., 2020). These findings cannot be explained by counting efficiency curves being dominated by diffusional losses (s. Fig. S5).

The figure clearly illustrates that composition-dependent differences in the 50 % cutoff diameters are steadily decreasing as a function of $S_{max}$. This statement is true for both the butanol-based and propylene glycol-based CPCs. In fair approximation the composition-dependence in the 50 % cutoff diameters vanishes at a peak saturation ratio of 4.6 and 5.0, for butanol and propylene glycol respectively. The figure suggests that there is a working fluid-specific saturation ratio at which the

composition dependence of particle activation vanishes. This is in line with the findings of Giechaskiel et al. (2011), who showed that activation efficiencies derived from heterogeneous nucleation theory, are expected to exhibit successively smaller differences for various seed material-vapor interactions (characterized by different microscopic contact angles) as the temperature difference inside of a CPC increases. While further completion of the figure by adding water- and DEG-based CPCs and boosters exceeds the scope of the presented study, this matter shall be urgently addressed in subsequent research

and must be taken into consideration when developing future CPC models.

## 4 Conclusions

In this study, we characterized the newly-developed Airmodus A30 using butan-1-ol and propane-1,2-diol as working fluids with four types of chemically different and size-selected seed particles under controlled laboratory conditions. We have

obtained comprehensive results on the composition dependence of the counting efficiency of the aforementioned CPC model. Our study highlights, that the propylene glycol-based A30 at default temperature settings does not exhibit a composition dependence of the cutoff diameters for NaCl, Ag, $(NH_4)_2SO_4$ and $C_{15}H_{24}O_x$ seeds. Propylene glycol has proven to be a valuable addition to the existing set of working fluids and the non-hazardous nature of the substance bears major benefits for studies of indoor air quality. Comparison to previous results and decreasing the saturator temperatures of the propylene glycol-based

model confirmed that a composition dependence of the described kind is mainly linked to the supersaturation profile seeds are exposed to. By lowering peak saturation ratios, we were able to re-induce a composition dependence of the counting efficiency in case of the propylene glycol-based CPC. The existence of a working fluid-specific supersaturation profile associated with no significant composition dependence of the counting efficiency should be validated by inclusion of other working fluids, water and DEG in particular, into the dataset. Notwithstanding the above, we are confident that our results will assist in the

future development of high-accuracy CPCs. Working fluids could be chosen such that very high supersaturations are necessary for a certain target $d_{50}$, thereby enabling e.g. a battery of CPCs (Kulmala et al., 2007) with different working fluids but all providing composition-independent cutoffs.

**Data availability:** The data set is available upon reasonable request addressed to the corresponding author.




**Author contributions:** DS conceptualized the study. PJW and DS performed the experiments, JE performed the CFD simulations, PJW and DS wrote the manuscript, PJW, JE, JV, AP, HG, PMW and DS were involved in the scientific discussion and interpretation of the results and commented on the manuscript.

**Competing Interests:** JE, JV and AP work for Airmodus Ltd. which sells the Airmodus A30 investigated in this paper. All other authors declare that they have no conflict of interest.

**Acknowledgements:** The authors thank Christan Tauber for valuable discussions. This research has been funded by the Vienna Science and Technology Fund (WWTF) through project VRG22-003 and by the Austrian Research Promotion Agency (FFG)
through BRIDGE project 39413939.





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





| CPC Model | Working Fluid | $T_{sat}$ [°C] | $T_{con}$ [°C] | $S_{max}$ [ ] | Material | $d_{50}$ [nm] | $\Delta d_{50}$ [nm] |
|---|---|---|---|---|---|---|---|
| A20 (default) | n-Butanol | 39 | 15 | 2.56 | Ag | 6.6 | 0.8 |
| | | | | | $(NH_4)_2SO_4$ | 6.8 | 0.8 |
| | | | | | $C_{15}H_{24}O_x$ | 6.0 | 0.7 |
| | | | | | NaCl | 10.4 | 1.2 |
| A30 (default) | Propylene Glycol | 48 | 12 | 5.00 | Ag | 5.6 | 0.7 |
| | | | | | $(NH_4)_2SO_4$ | 6.0 | 0.7 |
| | | | | | $C_{15}H_{24}O_x$ | 5.9 | 0.7 |
| | | | | | NaCl | 5.6 | 0.7 |
| A30 | Propylene Glycol | 41 | 12 | 3.56 | Ag | 6.0 | 0.7 |
| | | | | | $(NH_4)_2SO_4$ | 6.9 | 0.8 |
| | | | | | $C_{15}H_{24}O_x$ | 6.6 | 0.8 |
| | | | | | NaCl | 8.3 | 1.0 |
| A30 | Propylene Glycol | 37 | 12 | 2.93 | Ag | 8.9 | 1.1 |
| | | | | | $(NH_4)_2SO_4$ | 10.7 | 1.3 |
| | | | | | $C_{15}H_{24}O_x$ | 12.3 | 1.5 |
| | | | | | NaCl | 13.1 | 1.6 |

**Table 1:** Summary of 50 % cutoff diameters of the A30. The table summarizes the calculated 50 % cutoff diameters for the entire set of measurements and relates it to the temperature settings of saturator ($T_{sat}$) and condenser ($T_{con}$) of the CPC in use as well as to the calculated maximum saturation ratio $S_{max}$.


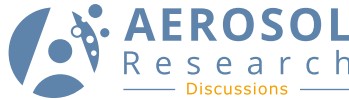



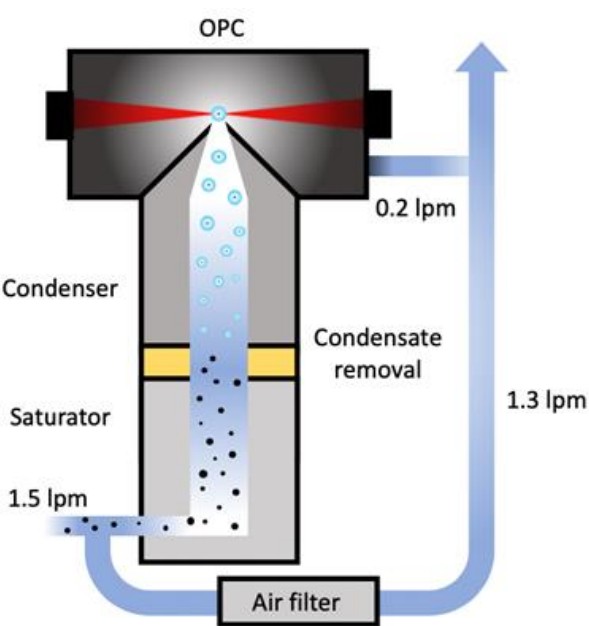

**Figure 1**: Schematic view of the Airmodus A30 condensation particle counter.









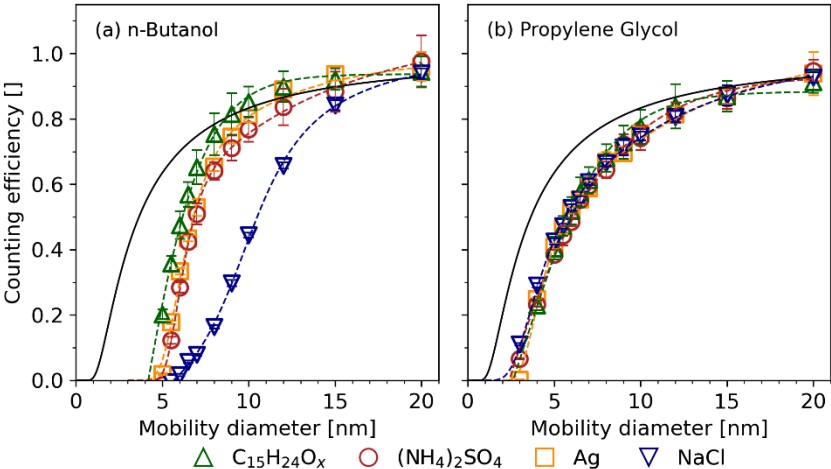

**Figure 2**: Counting efficiencies for butanol and propylene glycol. The figure presents the measured counting efficiencies as function of the mobility equivalent diameters. Panel (a) shows data measured with the butanol-based A30; panel (b) presents the results for the propylene glycol-based A30. Both CPCs were operated at default temperature settings (s. Table 1). The depicted markers correspond to different seed particles. The solid black line depicts the particles' transmission efficiency according to Gormley and Kennedy (1949).








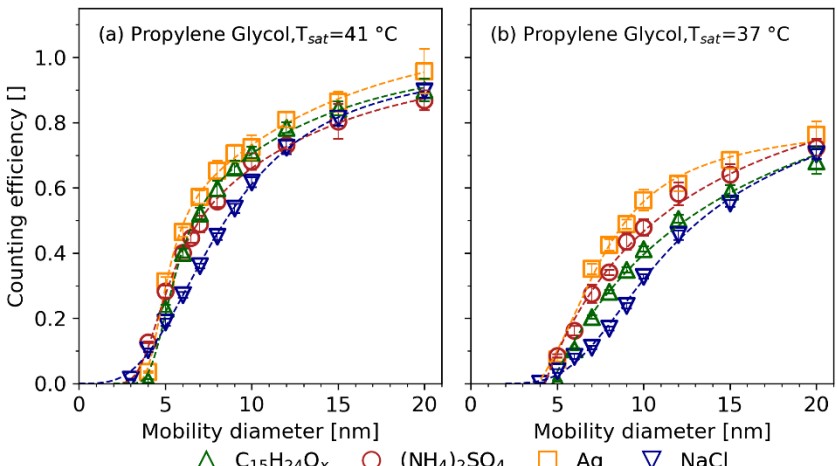

**Figure 3:** Counting efficiencies for decreased saturator temperatures. The figure presents the measured counting efficiencies as function of the mobility equivalent diameters measured with the propylene glycol-based A30. Saturator temperature ($T_{sat}$) was successively decreased prior to the measurements. Data presented in panel (a) corresponds to $T_{sat}$=41 °C; panel (b) shows the results for a saturator temperature of 37 °C. Markers correspond to different types of seed particles.







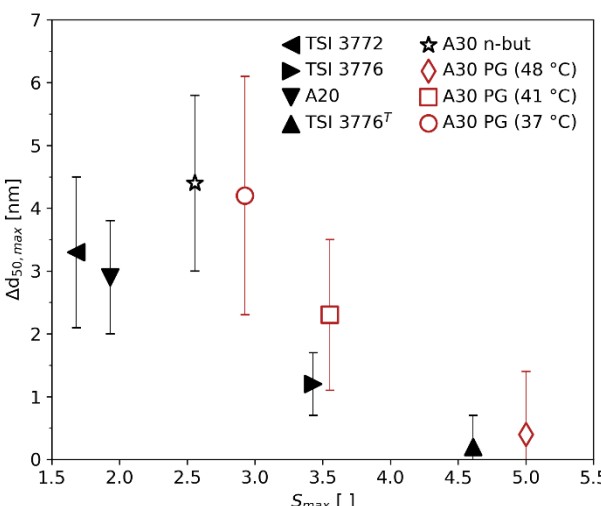

**Figure 4**: Maximum differences in 50 % cutoff diameters. The figure presents the difference between the minimum and maximum 50 % cutoff diameters for the seeds in use ($\Delta d_{50,max}$) as a function of the maximum saturation ratio. Data corresponding to filled markers were taken from Wlasits et al. (2020). The red markers correspond to data measured with the propylene glycol-based A30; black markers correspond to butanol-based CPCs. In case of the propylene glycol-based A30 the saturator temperatures are presented in brackets.
