# Peer review of "Reduced particle composition dependence in condensation particle counters"

_Aerosol Research, 2024_

## Author Comment (AC1)

**Authors' Response to Referee #1**

We thank Referee #1 for the critical assessment of our work and the helpful comments. In the following we address the comments point by point. Our responses are written in blue; the corresponding changes to the manuscript are presented in red:

*This well written manuscript describes the characterization of a newly-developed CPC model (A30, Airmodus Oy, Helsinki, Finland) for its operation with two different working fluids. The authors tested this CPC for its performance when operated with butanol and propylene glycol. The study was conducted for four different particle materials (sodium chloride, silver, ammonium sulfate and caryophyllene, which is a plant volatile found in essential oils). The authors found an influence of the particle materials on the counting efficiency and lower cut-off size of the CPC operated with butanol, while they did not observe any similar dependence of the CPC operated with propylene glycol. They then came to the somewhat hasty conclusion that the observed effect is only due to the working fluid, while they do not seem to consider that e.g. the very different operational temperatures used inside the CPC will also have played an important role. This is especially surprising as they later on decrease the saturator temperature of the propylene glycol-based CPC, but did not try to increase it in the butanol-based A30. The latter could have revealed if this new A30 CPC is possibly operated below its optimum supersaturation when using butanol as its working fluid. That said, the final conclusion is very sound and useful: "The existence of a working fluid-specific supersaturation profile associated with no significant composition dependence of the counting efficiency should be validated by inclusion of other working fluids".*

*Main comments:*

- *General comment: I am personally struggling with the term "seed material", which seems to say that seeds are to be measured by a CPC and possibly even imply that they are all of one and the same material. Instead, I would suggest to consider something along the lines of the "material dependence of particles acting as nuclei in a condensation particle counter".*

We agree with the reviewer that the phrase "seed material" is not precise enough in the context of our work. To avoid any ambiguity, we carefully reassessed the choice of words and amended the manuscript as follows:

Line 1: Reduced particle composition dependence in condensation particle counters

Line 46: However, for all of these CPCs it was recently shown that the cutoff curve is significantly influenced by the chemical composition of the seeds with differences in the $d_{50}$ values of up to 1 nm for the latest generation of ultrafine detectors (Wlasits et al., 2020).

Line 50: This implies a significant source of uncertainty within the typical applications of CPCs as the chemical composition of the sampled aerosol is often unknown and thousands of chemical species are distributed in different proportions across individual particles (Bondy et al., 2018).

Line 61: Several studies have shown that changes in the particle chemical composition also heavily influence these CPC measurements with a nominal cutoff of 23 nm posing an important challenge to achieve intercomparability of measurements within these legislative procedures (Giechaskiel et al., 2009; Giechaskiel et al., 2011; Terres et al., 2021; Krasa et al., 2023).

Line 65 : The development of CPCs with a reduced dependence on the chemical composition of the seed particles can thus be considered a milestone in aerosol technology. In the following sections we present conclusive evidence that aforementioned composition dependences can be efficiently reduced by using propylene glycol as condensable vapor in a newly developed CPC and that this behavior is related to high saturation ratios achieved in the instrument.

Line 238: Bondy, A. L., Bonanno, D., Moffet, R. C., Wang, B., Laskin, A. and Ault, A. P.: The diverse chemical mixing state of aerosol particles in the southeastern United States, Atmos. Chem. Phys., 18, 16, 12595-12612, https://doi.org/10.5194/acp-18-12595-2018, 2018.

- *Line 9: The authors state that "the composition dependence of particle activation is a source of profound uncertainty for the accurate assessment of particle number concentrations", which is too general a statement the way it is phrased. Apart from the measurement of nucleation events in ambient studies, the number of particles near a CPC's cut-off that either get or do not get activated and grow to be detected is typically a*

*relatively small fraction of the total number concentration of particles that is measured over the complete size range of a CPC.*

We agree with the reviewer that our statement was phrased in a very general way, especially when stated in the Abstract. As also pointed out by the reviewer, the implications of the composition dependence of particle activation are explained using new particle formation events in the Introduction (s. Lines 53-58). Consequently, we have restricted our statement to new particle formation events.

Line 10: Relying on heterogeneous nucleation as basic principle, the composition dependence of particle activation is a source of profound uncertainty for the accurate assessment of new particle formation (NPF) events.

- *Line 15: If "innovative choice of the working fluid" refers to propylene glycol then it should be made clear that propylene glycol or IUPAC has already been used in CPCs well over three decades ago (Sem, 2002). It was actually also used in a commercial CPC many years ago, e.g. the Model 3851 manufactured by KANOMAX that was introduced in 1986 (see McMurry, 2000).*

We thank the reviewer for the overview about the usage of propylene glycol as working fluid in previous CPC models. This fact has been made clear using the provided references. Consequently, we rephrased the associated passages of the manuscript:

Line 15: Our results conclusively demonstrate that composition-dependent particle detection can be reduced to the brink of disappearance by choice of the working fluid and corresponding high supersaturation.

Line 78: Consequently, propylene glycol was introduced as working fluid in a commercial CPC more than 25 years ago (Suzuki and Tarutani, 1997; McMurry, 2000; Sem, 2002). However, the influence of the chemical composition of the seed particles on the detection efficiency had yet to be determined.

Line 208: Propylene glycol remains to be a valuable addition to the existing set of working fluids and the non-hazardous nature of the substance bears major benefits for studies of indoor air quality.

Line 288: McMurry, P. H.: The History of Condensation Nucleus Counters, Aerosol Sci Technol., 33, 297-322, https://doi.org/10.1080/02786820050121512, 2000.

Line 295: Sem, G. J.: Design and performance characteristics of three continuous-flow condensation particle counters: a summary, Atmos. Res., 62, 3-4, https://doi.org/10.1016/S0169-8095(02)00014-5, 2002.

Line 305: Suzuki, I. and Tarutani, K.: Comparison of Sampling Techniques for the Analysis of Particulate Metal Impurities in Gases, Anal Sci, 13, 833-836, 1997.

- *Line 115: "These conspicuous discrepancies between the two identically constructed CPCs must be attributed to changes related to the working fluids in use". I am not convinced this is the (only) explanation. While the two A30 CPCs are "identically constructed", the authors previously stated that "propylene glycol-based CPCs (are) operated at a higher saturator temperature and a decreased condenser temperature compared to the butanol-based models". Tab. 2 shows a delta T of 36C for the IUPAC-A30, while that's only 24C for the butanol-A30. I could well imagine that the newly developed A30 CPC simply does not operate with optimized supersaturation conditions for butanol (and would also benefit from e.g. a higher saturator temperature). In particular as the data from the well-characterized butanol-based CPC 3776 in Fig. S5 (in the supplement) show no visible difference for the exact same particle materials.*

We fully agree with the referee. Indeed, Figure 4 suggests that the butanol-based A30 might not be operated at optimized supersaturation conditions. Consequently, we amended the manuscript as follows:

Line 122: We hypothesize that these conspicuous discrepancies between the CPCs derive from a combination of the following two conditions: 1) non-optimized supersaturation conditions in the butanol-based CPC; 2) more favorable interactions between vapor molecules and seed particles in case of the propylene glycol-based CPC.

Line 195: In addition, direct comparison of the butanol-based A30 to the propylene glycol-based A30 reveals further evidence for non-optimized supersaturation conditions in case of butanol.

- *Line 121: "These calculations revealed that the measured counting efficiencies cannot solely be attributed to diffusional losses within the instrument". This is somewhat poorly phrased and hard to understand. Why would the "measured counting efficiencies… be attributed to diffusional losses" in the first place? Is there something missing here?*

Following Stolzenburg and McMurry (1991), the counting efficiency of a CPC $\eta_{CPC}$ at a certain particle diameter $d_p$ is given by:

$$\eta_{CPC}(d_p) = \eta_{sam}(d_p) * \eta_{act}(d_p) * \eta_{det}(d_p),$$

where $\eta_{sam}$ refers to the sampling efficiency, $\eta_{act}$ is the fraction of particles activated in the condenser and $\eta_{det}$ denotes the fraction of particles detected in the optics block of the CPC. Hence, the counting efficiency as measured by our experimental setup is influenced by diffusional losses in the sampling system. We have calculated these losses (s. Fig. 2) and compared the curve shape to our experimental results. As a consequence of that comparison, we concluded that such diffusional losses are not solely responsible for the shape of the experimental curves. In first approximation we assumed a detection efficiency of 1. Hence, the experimental curve shapes must be influenced by the sampling efficiency but also by the activation efficiency.

To clarify, we amended the manuscript as follows:

Line 129: Following Stolzenburg and McMurry (1991), the counting efficiency of a CPC ($\eta_{CPC}$, the assessed quantity by our experimental setup) at a certain particle diameter ($d_p$) is given by

$$\eta_{CPC}(d_p) = \eta_{sam}(d_p) \cdot \eta_{act}(d_p) \cdot \eta_{det}(d_p), \tag{1}$$

where $\eta_{sam}$ refers to the sampling efficiency, $\eta_{act}$ is the fraction of particles activated in the condenser and $\eta_{det}$ denotes the fraction of particles detected in the optics block of the CPC. Since the counting efficiency is influenced by losses in the sampling system, diffusional losses along the particles' path from the CPC inlet to the condenser were assessed (solid lines in Fig. 2).

Line 136: These calculations revealed that the shape of the measured counting efficiencies cannot solely arise from diffusional losses within the instrument, i.e. the propylene glycol-based A30 does not lose its composition dependence just because it achieves unity activation efficiency for all particles.

- *Line 174: "The figure suggests that there is a working fluid-specific saturation ratio at which the composition dependence of particle activation vanishes." Yes, this is getting to the core of it!*
- *Line 178: "While further completion of the figure by adding water- and DEG-based CPCs and boosters exceeds the scope of the presented study, this matter shall be urgently addressed in subsequent research". Why limit this interesting sentence explicitly to water- and DEG-based CPCs? There are a lot of isopropanol CPCs in use and new substances such as dimethyl sulfoxide (DMSO) are increasingly investigated (Weber et al., 2023). Incidentally, the latter paper here in the AR journal also looked at the saturation ratio necessary to activate a sodium chloride particle of a given initial size for three working fluids: water, butanol and the new working fluid DMSO.*

We agree with the referee that the list of working fluids should be expanded. Hence, we have included isopropanol as well as DMSO in our statement:

Line 197: While further completion of the figure by adding water-, isopropanol- and DEG-based CPCs and boosters exceeds the scope of the presented study, this matter shall be urgently addressed in subsequent research and must be taken into consideration when developing future CPC models. In particular, the inclusion of more recently investigated and non-alcohol-based working fluids, like dimethyl sulfoxide (Weber et al., 2023), would yield further insights into the composition dependence of the counting efficiency in CPCs.

Line 321: Weber, P., Bischof, O. F., Fischer, B., Berg, M., Schmitt, J., Steiner, G., Keck, L., Petzold, A. and Bundke, U.: A new working fluid for condensation particle counters for use in sensitive working environments, Aerosol Res., 1, 1, 1-12, https://doi.org/10.5194/ar-1-1-2023, 2023.

*Minor Comments*

- *Line 86: "The overall design of the Airmodus A30 is presented in Fig. 1". This is the first time that instrument is mentioned, hence this is where the manufacturer's information should be shown.*

The manuscript was amended accordingly:

Line 92: The overall design of the Airmodus A30 (Airmodus Ltd., Helsinki, Finland) is presented in Fig. 1.

Consequently, the manufacturer information was removed from Line 100:

Line 100: Measurements were conducted using two versions of the newly developed Airmodus A30; i.e. the butanol-based A30 and the A30 operated with propylene glycol as working fluid, which are completely identical except for the used working fluid and temperature settings in saturator and condenser.

- *Line 95: "butanol-based variant and a variant of the A30 using propylene glycol". The use of the term variant is inconsistent with the later statement that the actual instruments are identical. I believe it is better to say something like "the butanol-based A30 and the A30 operated with propylene glycol".*

The sentence has been rephrased accordingly.

Line 100: Measurements were conducted using two versions of the newly developed Airmodus A30; i.e. the butanol-based A30 and the A30 operated with propylene glycol as working fluid, which are completely identical except for the used working fluid and temperature settings in saturator and condenser.

- *Line 165: A part of this sentence that says "the set of butanol-based CPC manufactured by TSI Inc." is a bit awkwardly phrased and the location details of the manufacturer are missing.*

The sentence has been split into two parts, rephrased, and the location details of the manufacturer were added.

Line 182: Data for this comparison were taken from Wlasits et al. (2020). The authors performed their measurements using a set of butanol-based CPCs (Model 3772 and Model 3776, TSI Inc., Shoreview, United States of America) including a tuned version of the TSI 3776, as well as using the Airmodus A20.

*References*

*McMurry, P. H. The History of Condensation Nucleus Counters, Aerosol Sci. Technol., 33:297–322, 2000*

*Sem, G. J. Design and performance characteristics of three continuous-flow condensation particle counters: a summary, Atmos. Res., 62: 3-4, 267-294, 2002*

*Weber, P., et al. A New Working Fluid for Condensation Particle Counters for Use in Sensitive Working Environments. Aerosol Res. 1, 1-12, 2023*

Additional Changes to the Manuscript

- Table 1/Row 2: "A20" replaced by "A30".
- Line 93: Blank spaces removed.
- Line 204: Hyphen removed.
- Line 229: Acknowledgements expanded: "Furthermore, the authors thank TU Wien and BMBWF for the purchase of the propylene glycol-based Airmodus A30."
- Line 315: Typo removed.
- Line 317: Typo removed.

**Authors' Response to Referee #2**

We thank Referee #2 for the critical assessment of our work and the helpful comments. In the following we address the comments point by point. Our responses are written in blue; the corresponding changes to the manuscript are presented in red:

*The authors present a study on a newly developed Airmodus A30 CPC, which is the latest model of the manufacturer to detect aerosol particle concentration even below the size range of 200 nm, where classical optical detection method of particles is not possible. The tested CPC's used 2 different working fluids namely the n-butanol and non-hazardous propylene glycol. During the experiments 4 different seed materials were applied (sodium chloride, silver, ammonium sulfate and caryophyllene). They experienced that the butanol-based CPC operated with the default saturator and condenser temperature had a strong dependence on the seed material, since a profound difference was measured between the counting efficiencies of different seed particles. During the experiments with the propylene glycol-based instrument no seed material dependence was observed at the default saturator temperature. When the saturator temperature of the propylene glycol-based CPC was decreased, the counting efficiency was decreasing as well. I fully agree with the other reviewer that it could have been interesting to modify the saturator temperature also in case of the butanol-based instrument. The results of this work could be very useful even for the manufacturers, regarding how to improve the performance of the newly developed instruments.*

*Main comments:*

- *Line 13: "Our results conclusively demonstrate that composition-dependent particle detection can be reduced to the brink of disappearance by innovative choice of the working fluid.". Line 189: "Comparison to previous results and decreasing the saturator temperatures of the propylene glycol-based model confirmed that a composition dependence of the described kind is mainly linked to the supersaturation profile seeds are exposed to. By lowering peak saturation ratios, we were able to re-induce a composition dependence of the counting efficiency in case of the propylene glycol-based CPC". Please clear, which is the main outcome of the work, whether the innovative choice of the working fluid, or dependence on the supersaturation profile seeds are exposed or both.*

Subsequent to the comments of Referee #1 outlining the previous use of propylene glycol as working fluid for CPCs many years ago, we have clarified the outcome of our study. Consequently, the proposal of a working fluid-specific saturation ratio, at which the composition dependence of particle activation vanishes, is the main outcome of our work (s. Fig. 4). In that context, we would like to point the reviewer's attention to the following clarifying changes:

Line 15: Our results conclusively demonstrate that composition-dependent particle detection can be reduced to the brink of disappearance by choice of the working fluid and corresponding high supersaturation.

Line 78: Consequently, propylene glycol was introduced as working fluid in a commercial CPC more than 25 years ago (Suzuki and Tarutani, 1997; McMurry, 2000; Sem, 2002). However, the influence of the chemical composition of the seed particles on the detection efficiency had yet to be determined.

Line 208: Propylene glycol remains to be a valuable addition to the existing set of working fluids and the non-hazardous nature of the substance bears major benefits for studies of indoor air quality.

Line 288: McMurry, P. H.: The History of Condensation Nucleus Counters, Aerosol Sci Technol., 33, 297-322, https://doi.org/10.1080/02786820050121512, 2000.

Line 295: Sem, G. J.: Design and performance characteristics of three continuous-flow condensation particle counters: a summary, Atmos. Res., 62, 3-4, https://doi.org/10.1016/S0169-8095(02)00014-5, 2002.

Line 305: Suzuki, I. and Tarutani, K.: Comparison of Sampling Techniques for the Analysis of Particulate Metal Impurities in Gases, Anal Sci, 13, 833-836, 1997.

*Minor Comments*

- *Line 13: The first time Airmodus A30 mentioned. Please put the manufacturer information as in line 95 written.*

The manuscript was amended accordingly:

Line 13: Addressing this pressing problem, we present calibrations of a newly developed CPC, the Airmodus A30 (Airmodus Ltd., Helsinki, Finland), that uses non-hazardous propylene glycol as working fluid.

Additional Changes to the Manuscript

- Table 1/Row 2: "A20" replaced by "A30".
- Line 93: Blank spaces removed.
- Line 204: Hyphen removed.
- Line 229: Acknowledgements expanded: "Furthermore, the authors thank TU Wien and BMBWF for the purchase of the propylene glycol-based Airmodus A30."
- Line 315: Typo removed.
- Line 317: Typo removed.